# Self-care practice and its barriers among diabetes patients in North East Ethiopia: A facility-based cross-sectional study

**Adisu Asefa** [1] *, **Abebe Muche Belete**[2], **Feredegn Talarge**[2], **Daniel Molla**[2]

1 Communicable and Non-Communicable Disease Directorate, Armauer Hansen Research Institute, Addis Ababa, Ethiopia, 2 Department of Biomedical Science, College of Medicine, Debre Berhan University, Debre Berhan, Ethiopia

* sadamasefadb@gmail.com

## Abstract

Diabetes prevention and management through self-care practice is critical to reducing severe complications and death due to diabetes. Data on the prevalence of self-care practices will help us to design and implement prevention and management strategies to foster adherence and compliance with the interventions. This study was intended to assess self-care preparation and its barriers among diabetes patients in Northeast Ethiopia. A facility-based cross-sectional study was conducted among diabetes patients visiting Debre Berhan Town Public Health Institutions from March 10, 2021- April 10, 2021. A systematic random sampling technique was utilized to select 392 samples. Data were collected using a structured questionnaire adapted from Summary of Diabetes Self-Care Activities Measures. Reliability analysis was done using Cronbach's alpha test, and the Hosmer and Lemeshow test also checked for model fitness. Bivariate and multivariable binary logistic regression was done to identify the factors associated with dietary practices. For all statistically significant tests, p- a value < 0.05 was used as a cut-off point. The mean age of the respondents was 47.1 years, with a standard deviation (SD) of ± 13.4 years. The mean adherence to self-care practice was 29.00 ± 10.37 SD. More than half, 218 (61.1%) of the study subjects had poor self-care practices. In the multivariable logistic regression analysis, being a governmental worker (AOR = 7.06 (1.61–30.9) and having social support from partners (AOR = 5.83 (3.01–11.3) showed a statistically significant association with good self-care practice. The current study showed that the overall level of self-care practice of study subjects was poor. Therefore, health facilities should provide adequate health education and promotion activities to enhance patients' level of adherence. In addition, families, partners, or friends of diabetes patients should be informed about their essential roles in patients' self-care practice.

**Data Availability Statement:** The English and Amharic version questionnaires used for my data collection were uploaded as supplementary

information. The questionnaire alone is sufficient for the replication of this study.

**Funding:** The authors received no specific funding for this work.

**Competing interests:** The authors have declared that no competing interests exist.

## Introduction

In most developed countries, diabetes is the fourth or fifth leading cause of death, and increasing evidence shows that it has reached epidemic proportions in many developing countries [1, 2].

Type 2 diabetes results from the interaction between genetic capacity, high-risk behaviours, and environmental risk factors [3]. According to the 2019 International Diabetes Federation (IDF) report, approximately 463 million adults (20–79 years) have diabetes; by 2045, this number will increase to 700 million [4].

Ethiopia is one of the sub-Saharan countries, and the double burden of its population is increasing epidemiological risks, namely, cancer, cardiovascular diseases, chronic respiratory diseases, and diabetes, which cause 60% of global deaths but receive just 3% of international development assistance for health [5].

The foundation of diabetes management is adherence to lifestyle interventions, including healthy eating, regular physical exercise, smoking cessation, and maintaining a healthy weight [6]. Diabetes-related complications and mortality are prevented and controlled primarily by adopting healthy lifestyle habits. However, this is a challenging problem in developing countries where the quality of healthcare services could be better [7].

Regardless of the type, 95% of diabetes control is based on diet and lifestyle interventions, and the patient or his family usually provides 95% of self-care. Therefore, diabetic patients must adjust their behaviour, such as changing their lifestyles to change their diet and physical activity level and following prescribed treatments to prevent diabetes complications, which can be fatal, especially for people [8–12].

A large percentage of chronic illnesses are prevented and managed through the reduction of the four main shared behavioural risk factors: tobacco use, physical inactivity, the harmful use of alcohol, and unhealthy diets (referred to as modifiable risk factors). For example, up to 80% of heart disease, stroke, and type 2 diabetes and about 40% of cancers could be prevented by controlling these risk factors [5].

Evidence from a systematic review and meta-analysis study conducted in Ethiopia showed that the pooled prevalence of good diabetes self-care practice among diabetic patients was 49%. This study characterized the effective regimen of diabetes self-care based on pooled prevalence. The pooled estimate for diet was 50%, self-blood glucose monitoring was 28%, recommended physical activity was 49%, and diabetic foot care was 58% [13].

A study conducted in Northern Ethiopia revealed that the overall good self-care practice was 50.3%. This study also showed that 39.3% and 54.9% of the study participants had good exercise and diet adherence, respectively [14]. Another survey from Ethiopian General Hospital indicated that (63.1%) of study participants had good self-care practices. In addition, this study identified that patients who had a glucometer at home had social support and received a diabetes education, which was statistically associated with good self-care practice [15].

According to a study in Addis Ababa, Ethiopia, three-fourths (75.9%) of study patients did not follow the recommended diet management (and 83.5%) did not adhere to self-monitoring of blood glucose levels. In comparison, a few (4.3%) respondents did not take the administered diabetes medications [16].

However, more about self-care practice in North-East Ethiopia needs to be studied. To prevent complications related to diabetes, it is essential to find out how patients with type 2 diabetes adhere to self-care measures. Self-care practice is vital for health sector stakeholders to design and implement prevention strategies for diabetes-related complications; it also reduces medicine's direct and indirect costs. Therefore, the current study might help identify gaps in areas of self-care practice which are essential in many aspects, both for the individual's well-being and from a socioeconomic perspective [Fig 1].

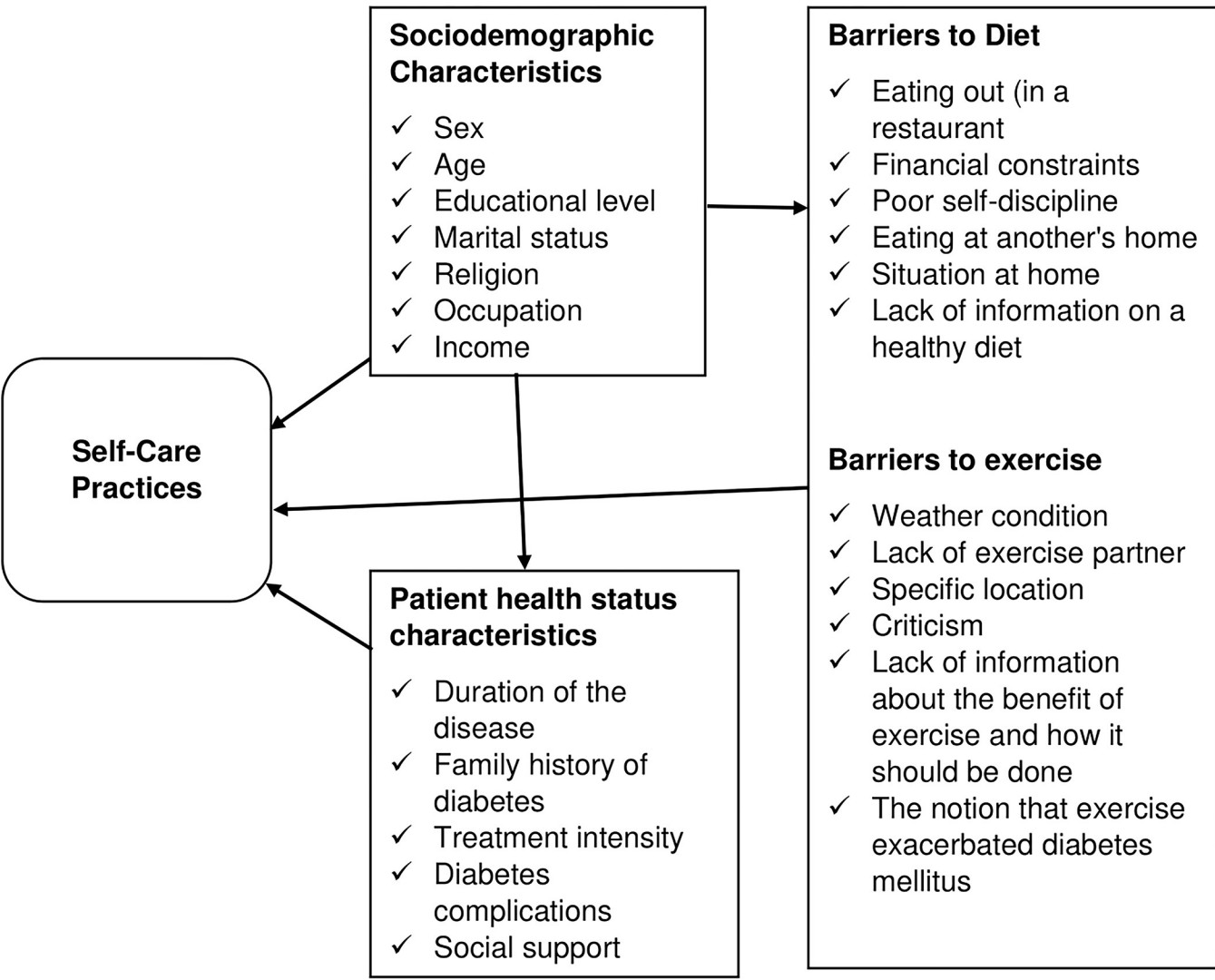

**Fig 1. Conceptual framework for studying barriers to self-care practices among diabetes patients [34].**

## Materials and methods

### Study design and setting

A facility-based cross-sectional study was carried out among diabetes patients attending Debre Berhan Town Public Health Institutions, North-East Ethiopia, from March 10, 2021- April 10, 2021.

### Sample size determination

According to a previous study conducted in Ethiopian teaching hospitals, adherence to dietary recommendations was 44.3% [17]. Therefore, the minimum sample size required for this study was determined using the single population proportion formula.

(ni = (Zα/2)2 p (1-p) /d2), Taking p = 44.3%, a 5% precision (d) level with a 95% confidence interval plus a 10% non-response rate was added. A population correction formula was also utilized. The final sample size was = 392.

### Sampling procedures and techniques

Four public health institutions in Debre Berhan town, North-East Ethiopia, were included in the study. Based on the count from the registration books on diabetic follow-up clinics, the average number of diabetes attending the selected institutions per month was 815.

A systematic random sampling technique was used among 815 patients coming to these institutions for a follow-up service. The sampling fraction was: 815/392 = 2.1. The first sample was selected by using a simple random sampling technique. Then, every two patients from each public health institution were included in the study until the calculated sample size was achieved [Fig 2].

### Study variables

**Dependent variable.** Self-care practice.

### Independent variables

**Sociodemographic variables.** Age, sex, marital status, educational status, occupation, and income

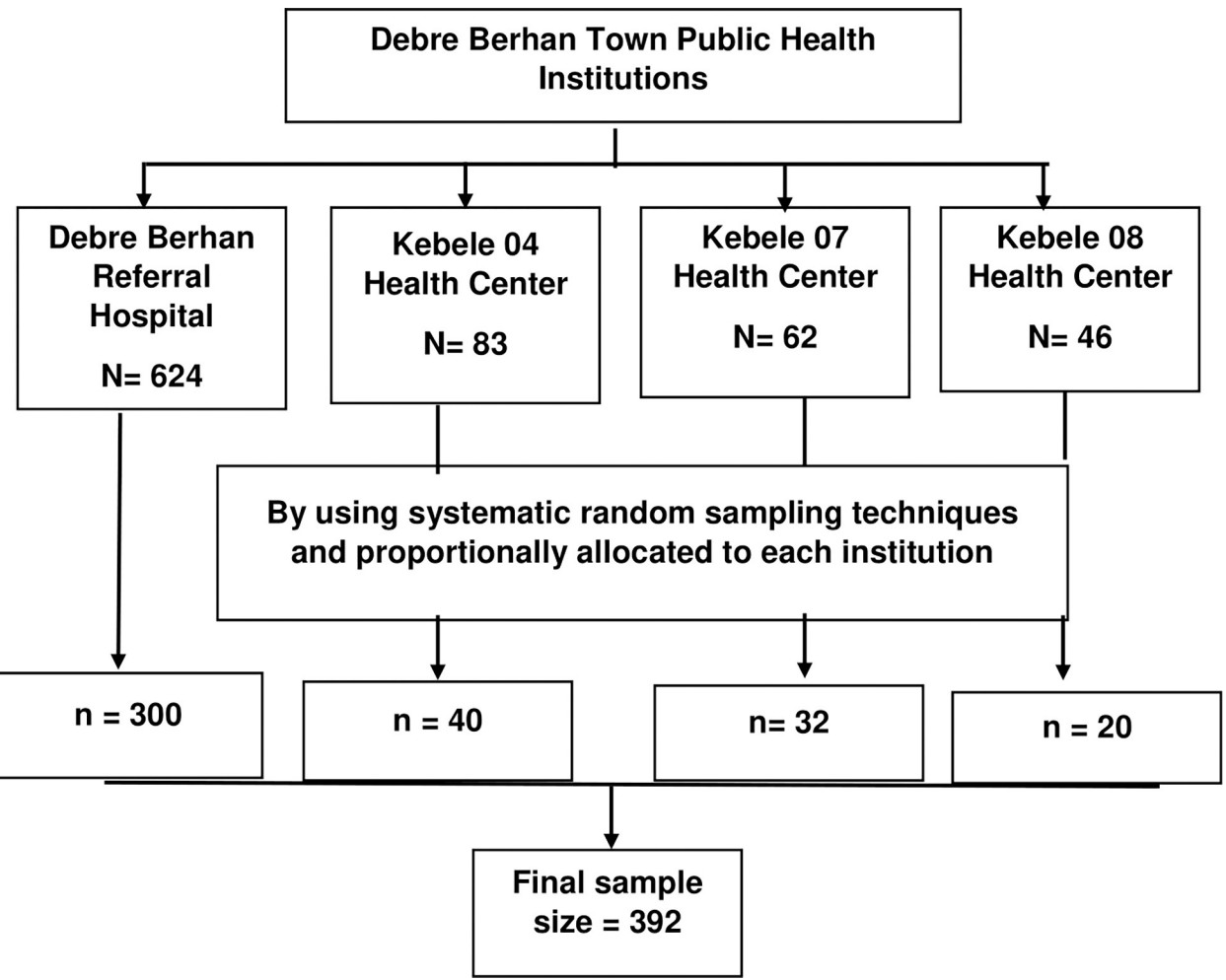

**Fig 2. Flow chart indicating sampling procedure in Debre Berhan Town Public Health Institutions.**

**Patient health status variables.** Duration of the disease, family history of diabetes, treatment intensity, diabetes complications, social support, and cigarette smoking status.

## Data collection tools and procedures

Data were collected using a structured questionnaire adapted from the Summary of Diabetes Self-Care Activities Measure (SDSAM) [18]. The questionnaires consist of four parts: sociodemographic characteristics, patient health status, a summary of diabetes self-care activities measure, and barriers to lifestyle and dietary pattern interventions. Overview of Diabetes Self-Care Activities Measure (SDSAM) asked about the diabetes self-care activities during the past seven days. The Summary of Diabetes Self-Care Activities Measure (SDSAM) included five components: diet, exercise, blood sugar testing, foot care, and smoking. Scores were calculated for each of the five regimen areas assessed by the SDSCA: Diet, Exercise, Blood-Glucose Testing, Foot Care, and Smoking Status. The mean value of lifestyle and dietary practices for the study participants was calculated using a score ranging from 0 to 7 for each item of the five SDSCA domains. For general diet = the mean number of days for items 1 and 2, specific diet = a mean number of days for items 3 and 4, reversing item 4 (0 = 7, 1 = 6, 2 = 5, 3 = 4, 4 = 3, 5 = 2, 6 = 1, 7 = 0), exercise = a mean number of days for items 5 and 6, blood-glucose testing = a mean number of days for items 7 and 8, foot-Care = a mean number of days for items 9 and 10, smoking Status = Item 11 (0 = non-smoker, 1 = smoker), and several cigarettes smoked per day [18].

The reliability of the SDSCA questionnaire was tested using Cronbach's alpha test. The overall reliability was 0.757. The questionnaire was prepared in English, translated into the Amharic version, and then back to the English version. Study participant recruitment was conducted from March 10, 2021 to April 10, 2021. Five BSc health professionals were recruited to collect the data, and two BSc/ MSc health professionals supervised the data collection process. The training was given to the data collectors and supervisors for three days to maintain data quality. Timely supervision was carried out by the principal investigator daily.

## Data processing and analysis

The data was entered into EPI data manager version 3.3 and analyzed using IBM SPSS Statistics version 22. Reliability was examined using Cronbach's alpha test. Model fitness was also checked by the Hosmer and Lemeshow test. A summary of descriptive statistics was computed for most variables. Bivariate analysis was done to determine the presence of an association between lifestyle and dietary pattern intervention practice with other variables. To avoid unstable estimates in the subsequent model, only variables that reached a p-value less than 0.2 at bivariate analysis were included in the following model analysis. Multiple logistic regression analysis was applied to describe the functional independent barriers of adherence to self-care practice. A point estimate of Odds ratio (OR) with a 95% confidence interval (CI) was determined to assess the strength of association between independent and dependent variables. For all statistically significant tests, p- a value <0.05 was used as a cut-off point.

## Ethics statement

This study was conducted after ethical approval letters obtained from the Institute of Research Ethics and the review board (IRB) of the College of Health Science, Debre Berhan University, Ethiopia, complied with the Declaration of Helsinki. First, the Institute of Research and Ethics Board Committee (IRB) reviewed and looked at the study's originality, feasibility, and ethical aspects. Following thorough discussion, the committee approved the research proposal by authors with ethical approval using reference number SPH/112/2020. Then, permission was taken from the hospital and health centers' higher management, and data were collected after

obtaining written informed consent from the study participants. To keep confidentiality, codes were used, and unauthorized persons did not have access to the data.

### Operational definition

**Self-care practice.** Refers to self-care activities such as following a diet plan, avoiding high-fat foods, regular physical activity, self-glucose monitoring, and administration of medication engaged for the past seven days, which is measured by Summary of Diabetes Self-Care Activities [18].

**Poor self-care practice.** Those patients' adherence to self-care behaviours scored below the mean of self-care behaviour items.

**Good self-care practice.** Those patients' adherence to self-care behaviours scored above the mean of self-care behaviour items.

**Family history of diabetes.** having a history of diabetes of their parents and first- and second-level relatives as self-reported.

**Lifestyle management.** non-pharmacological management such as physical exercise, foot care activities, and smoking cessation is designed to treat the problems of type 2 DM patients.

**Self-care behaviours mean score.** (Number of days patient practiced specific behaviour)/ (Total number of days under questions designed for that behaviour).

## Results

A total of 357 study participants were successfully involved in the study, yielding a response rate of 91.1%. The majority (61.1%), (71.7%), and (81%) of study participants were men, married, and able to read and write, respectively. Their ages ranged from 18 to 86 years, with a mean age of 47.1 ± 13.4 years. The average monthly income for the study participants was 4070 ETB. Most of the participants, 95 (26.6%), worked private jobs [Table 1].

### Health status characteristics of participants

The study result revealed that most 316 (88.5%) respondents had no family history of diabetes. Most (95.2%) of the respondents were non-smokers. Approximately three-fourths of the respondents (72%) have incurred diseases more significant than five years of duration. Of the total respondents, 296 (82.9%) had no diabetes-related complications. More than half (58.5%) of them had no social support (partner, family, friend, etc.) [Table 2].

### Self-care practices adherence

The frequency of respondents in a diabetes self-care behaviour defined by the number of days per week was analyzed by calculating the mean score, which refers to the average number of days respondents adhered to self-care measures in the previous seven days. Based on this, the overall mean and standard deviation of adherence to self-care practice were 29.00 ± 10.37 [Table 3].

The overall level of adherence to self-care practice was categorized as good and poor adherence using an average mean score. The current study revealed that 218 (61.1%) subjects had poor self-care practices. Two hundred six (57.7%) of the study subjects had poor dietary pattern adherence, while two hundred one (56.3%) had poor exercise adherence.

### Self-care recommendations for lifestyle and dietary pattern intervention practice

More than half (58.3%) of the study subjects were advised about eating low dietary fat in their meals. Similarly, 184 (51.5%) were advised about eating at least five servings of fruits and vegetables daily. However, 66 (18.5%) had yet to give any advice about their dietary pattern.

**Table 1. Sociodemographic characteristics of study participants, Northeast Ethiopia, 2021.**

| Sociodemographic Variables | Frequency (%) |
|---|---|
| **Sex** | |
| Male | 218 (61.1) |
| Female | 139 (38.9) |
| **Age group** | |
| 18–24 | 14 (3.9) |
| 25–64 | 306 (85.7) |
| > 64 | 37 (10.4) |
| **Marital status** | |
| Single | 53 (14.9) |
| Married | 256 (71.7) |
| Widowed | 24 (6.7) |
| Divorced | 24 (6.7) |
| **Occupation** | |
| Government worker | 55 (15.4) |
| Private worker | 180 (50.4) |
| Housewife | 67 (18.8) |
| Others | 55 (15.4) |
| **Educational status** | |
| Unable to read and write. | 68 (19) |
| Able to read and write. | 99 (27.7) |
| Primary school | 46 (12.9) |
| Secondary school | 69 (19.3) |
| College or University | 75 (21) |

**Table 2. Health status characteristics of study participants, Northeast Ethiopia, 2021.**

| Variables | Frequency (%) |
|---|---|
| **Duration of diseases** | |
| Less than one year | 36 (10.1) |
| From one to five years | 64 (17.9) |
| Greater than five years | 257 (72) |
| **Family history of diabetes** | |
| Yes | 41 (11.5) |
| No | 316 (88.5) |
| **Treatment intensity** | |
| Oral hypoglycemic agent | 283 (79.3) |
| Insulin therapy | 45 (12.6) |
| Combination | 29 (8.1) |
| **Social support** | |
| Yes | 148 (41.5) |
| No | 209 (58.5) |
| **Currently having glucometer** | |
| Yes | 32 (9) |
| No | 325 (91) |
| **Diabetes complication** | |
| Yes | 61 (17.1) |
| No | 296 (82.9) |
| **Cigarette smoking** | |
| Yes | 17 (4.8) |
| No | 340 (95.2) |

**Table 3. Mean score of self-care practices in Northeast Ethiopia, 2021.**

| Self-care measures (0–7 days) | Mean (SD) |
|---|---|
| **Overall self-care adherence** | **29.00 ± 10.37** |
| Dietary pattern | 16.47 ± 5.08 |
| General diet | **6.35 ± 3.66** |
| Specific diet | **10.12 ± 1.98** |
| Physical exercise | **4.78 ± 2.56** |
| Blood glucose | **3.39 ± 2.59** |
| Foot care | **4.35 ± 3.61** |

Three hundred two (84.6%) and 97 (27.2) of the respondents were advised regarding adherence to low-level physical exercise (such as walking) daily and the importance of participating in sports activities (specific amount, type, duration, and level of exercise), respectively. But 48 (13.4%) reported that they had not given any advice about physical exercise.

The health care team (doctor, nurse, and health officer) has also advised testing blood sugar levels using a drop of blood from a finger prick using a colour chart. Two hundred ninety-four (82.4%) reported that they were advised about the importance of testing blood sugar levels, but 54 (15.1%) said that they had yet to give any advice about blood sugar testing.

Two hundred seventy-one (75.9%) of them were advised on how to take diabetes pills to control blood sugar levels.

## Barriers to self-care practices

In order of magnitude, the stated barriers for poor dietary adherence included, (62.3%), (61.6%), (40.6%), (21.6%), and (15.7%) of the study subjects described poor self-discipline, difficult situations at home, financial constraint, lack of information about the list of healthy diets, eating out in a restaurant, and eating at the neighbourhood, family or other respectively.

In order of magnitude, the stated barriers to physical exercise adherence included, (79.6%), (30.5%), (and 11.5%) of the study subjects described icy weather conditions of the town, lack of information about the benefit of exercise and how it should be done, and the belief that physical activity exacerbates the needs of the disease respectively [Table 4].

## Factors associated with self-care practice

During the multivariable logistic regression analysis, government workers were 7.06 times more likely to have good self-care practices than other professions (AOR = 7.06; 95% CI: (1.61–30.9)). Concerning educational status, those study subjects who were unable to read and write, able to read and write, and attending primary school were 72.8%, 60%, 66%, respectively, less likely to have good self-care practice than those study subjects who attended college or university (AOR = 0.28; 95% CI: (0.09–0.83), AOR = 0.40; 95% CI (0.16–0.99), AOR = 0.34; 95% CI(0.12–0.98)), respectively.

Study subjects with social support from partners, family, and friends were 7.40 times more likely to have good self-care practices than those without family support (AOR = 7.40 (4.03–13.57)). Furthermore, study subjects who had a self-glucometer machine in their home were 5.82 times more likely to have good self-care practice than those who had no self-glucometer machine in their home (AOR = 5.82(1.83–18.4)) [see Table 5].

During the multivariable logistic regression analysis, those who were 15–24 years age group were 93% less likely to adhere to good dietary practices than those who were > 64 years age group. Furthermore, those with social support from partners, family, and friends were 5.83

**Table 4. Barriers to lifestyle and dietary intervention among diabetes in Northeast Ethiopia, 2021.**

| Barriers to dietary pattern and exercise adherence | Frequency (%) | |
|---|---|---|
| | Yes (%) | No (%) |
| **Dietary pattern** | | |
| Eating out (in a restaurant) | 63 (17.6%) | 294 (82.4%) |
| Financial constraints | 145 (40.6%) | 212 (59.4%) |
| Self-discipline | 222 (62.3%) | 135 (37.8%) |
| Eating at another's home | 56 (15.7%) | 301 (84.3%) |
| Difficult situation at home | 220 (61.6%) | 137 (38.4%) |
| Lack of information on a healthy diet | 77 (21.6%) | 280 (78.4%) |
| **Physical exercise** | | |
| The very cold weather condition of the town | 284 (79.6%) | 73 (20.4%) |
| Lack of exercise partner | 49 (13.7%) | 308 (86.35) |
| Lack of specific location | 107 (30%) | 250 (70%) |
| Criticism | 26 (7.3%) | 331 (92.7%) |
| Lack of information about the benefit of exercise | 109 (30.5%) | 248 (69.5%) |
| The belief that exercise exacerbated diabetes mellitus | 41 (11.5%) | 316 (88.5%) |
| Others | 8 (2.2%) | 349 (97.8%) |

times more likely to adhere to good dietary practices than those without family support (AOR = 5.83(3.01–11.3)). Finally, study subjects eating out in restaurants were 76% less likely to adhere to good dietary practices than those not eating in restaurants.

Male study subjects were 2.01 times more likely to adhere to good physical exercise practice than females (AOR = 2.01 (1.02–3.98)).

## Discussion

The majority of diabetic patients in our study lacked comprehensive self-care practice. Most of the study participants undermine the significance of the elements of self-care practice either due to lack of information, lack of resources/poverty, lack of social support or negligence.

The study showed that the overall mean and standard deviation of adherence to diabetes self-care practice measures was 29.00 ± 10.37. The current finding was lower than other West Shoa Zone studies, which were 39.8 ± 9.5 [19]. This difference might be due to differences in socioeconomic status. However, this finding was higher than a study conducted in public hospitals found in western Ethiopia, which was 23.09 ±6.55 [20]. Therefore, the high mean score for self-care practice indicates good adherence, which might significantly contribute to disease control. Furthermore, it reduces complications related to diseases.

The current study revealed that 61.1% of the subjects had poor self-care practices. Comparable findings were revealed in a study conducted at public hospitals in western Ethiopia, a study done at Jimma University Specialized Teaching Hospital and a study from Harari, of which 57.3%, 55%, and 60.7% of the participants had poor self-care practice, respectively [21–23]. Similar finding was noted in a study conducted in Jeddah, Saudi Arabia, which showed that 62% of participants had inadequate practice in self-management [24]. However, the current finding was higher than the finding from the systematic review and meta-analysis conducted by Dagnew et al., which was 48.88% [25] and the result of Chali et al., which was 45.7% [26]. This difference might be due to the scope of the study, socioeconomic disparities, and sample size variations. Similarly, this finding was lower than a multinational web-based survey conducted in the UK, US, Australia, and Japan. The level of self-care practice was highest in Japan (54.9%), followed by the UK (43.1%), the USA (42.5%), and Australia (40.4%) [27]. The

**Table 5. Factors associated with self-care practice among diabetes in Northeast Ethiopia, 2021.**

| Variables | COR (CI) | AOR (CI) | P-value |
|---|---|---|---|
| *Sociodemographic* | | | |
| **Age group** | | | |
| 15–24 | 2.06(0.48–8.82) | 0.27(0.03–1.97) | 0.19 |
| 25–64 | 3.76(1.52–9.29) | 0.89(0.25–3.21) | 0.86 |
| >64 | 1 | 1 | |
| **Marital status** | | | |
| Single | 3.36(1.15–9.79) | 3.10(0.76–12.60) | 0.11 |
| Married | 2.02(0.77–5.25) | 1.83(0.59–5.65) | 0.29 |
| Widowed | 0.27(0.04–1.51) | 0.50(0.07–3.46) | 0.48 |
| Divorced | 1 | 1 | |
| **Occupation** | | | |
| Private worker | 3.69 (1.70–7.98) | 1.92 (0.60–6.06) | 0.26 |
| Governmental worker | 17.6 (6.62–47.0) | 7.06 (1.61–30.9) | **0.01**[*] |
| Housewife | 1.35 (0.53–3.40) | 1.48 (0.44–4.98) | 0.52 |
| **Educational status** | 1 | 1 | |
| Unable to read and write | 0.07(0.03–0.17) | 0.28(0.09–0.83) | **0.02**[*] |
| Able to read and write | 0.14(0.07–0.28) | 0.40(0.16–0.99) | **0.04**[*] |
| Primary school | 0.12(0.05–0.29) | 0.34(0.12–0.98) | **0.04**[*] |
| Secondary school | 0.31(0.15–0.63) | 0.66(0.27–1.65) | 0.38 |
| College or university | 1 | 1 | |
| **Income category** | | | |
| Lower-income | 0.18(0.11–0.29) | 0.68(0.31–1.48) | 0.33 |
| Higher-income | 1 | 1 | |
| *Health status characteristics* | | | |
| **Duration of diseases** | | | |
| Less than one year | 2.20(1.08–4.46) | 1.77(0.63–4.99) | 0.27 |
| From one to five year | 1.20(0.68–2.11) | 1.60(0.73–3.47) | 0.23 |
| Greater than five years | 1 | 1 | |
| **Treatment intensity** | | | |
| Oral hypoglycemic agent | 0.52(0.24–1.13) | 3.29(1.10–9.77) | **0.03**[*] |
| Insulin therapy | 0.29(0.11–0.79) | 1.17(0.32–4.29) | 0.80 |
| Combination | 1 | 1 | |
| **Social support** | | | |
| Yes | 7.56(4.68–12.20) | 7.40(4.03–13.57) | **0.01**[*] |
| No | 1 | 1 | |
| **Currently glucometer** | 10.27(3.84–27.39) | 5.82(1.83–18.4) | **0.01**[*] |
| Yes | 1 | 1 | |
| No | | | |

[*]Statistically significant association

desire for self-reliance could explain this difference in participants from all four countries. The second common reason could be the cost-effectiveness of practicing self-care measures in these countries [27].

This study also showed that study subjects who were unable to read and write, able to read and write, and attending primary school were 72.8%, 60%, and 66%, respectively, less likely to have good self-care practice than those respondents who had college or university degrees.

Studies from other settings showed that diabetes patients with a higher level of educational status are engaged more in self-care practice [28, 29].

The current study revealed that (57.7%) of the study subjects had poor dietary pattern practice. Comparable findings were noted in a study conducted in Addis Ababa, which indicated 51.4% was an insufficient level of adherence [30].

The current finding revealed that social support group was one predictor factor that affects dietary pattern practice in diabetes patients; in this study, respondents who had social support from partners, family, and friends were 5.83 times more likely to adhere to good dietary practice than those who had lack of family support (AOR = 5.83(3.01–11.3)). This finding was supported by the study conducted in Benishangul Gumuz [26] and Jimma [31]. Social support from partners, family members, and friends positively predicts diet and exercise recommendations adherence. "Family and friends play both supportive and obstructive roles. Participants identified both positive and negative contributions of family and community members to their diabetes management [32]. Study subjects eating out in restaurants were 76% less likely to adhere to good dietary practices. These findings are comparable with a study conducted by Ayele et al. [33].

## Conclusion

The current study showed that the overall level of self-care practice was poor. Based on our findings, we recommend that health professionals working in Debre Berhan town public health institutions should provide adequate health education and promotion activities to enhance patient's level of adherence to self-care activities measures, and families, partners, or friends of diabetes patients should be informed about their essential roles in encouraging, reinforcing and supporting the patients for the better adherence of self-care activities measure.

## Supporting information

**S1 Annex. Questionnaire.**
(PDF)

## Acknowledgments

We want to thank Debre Berhan Referral Hospital, Kebele 04, 07, and 08 health facilities, for their kind cooperation and support. We also acknowledge Debre Berhan University for its kind collaboration in conducting the study.

## Author Contributions

**Conceptualization:** Adisu Asefa.

**Data curation:** Abebe Muche Belete.

**Formal analysis:** Adisu Asefa.

**Methodology:** Adisu Asefa.

**Software:** Adisu Asefa, Feredegn Talarge.

**Supervision:** Abebe Muche Belete.

**Visualization:** Feredegn Talarge, Daniel Molla.

**Writing – original draft:** Adisu Asefa.

**Writing – review & editing:** Adisu Asefa, Abebe Muche Belete, Feredegn Talarge, Daniel Molla.

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
