## [Decision Letter · Decision Letter 0]

25 Oct 2022

PGPH-D-22-00339

Self-Care Practices and its Barriers among Diabetes Patients in North East Ethiopia: A facility-based cross-sectional study

Dear Dr. Asefa,

Thank you for submitting your manuscript to PLOS Global Public Health. After careful consideration, we feel that it has merit but does not fully meet PLOS Global Public Health’s publication criteria as it currently stands. Therefore, we invite you to submit a revised version of the manuscript that addresses the points raised during the review process.

We look forward to receiving your revised manuscript.

Kind regards,

Lingkan Barua, MBBS, MPH

Academic Editor

Journal Requirements:

1. Please provide additional details regarding participant consent. In the ethics statement in the Methods and online submission information, please ensure that you have specified (1) whether consent was informed and (2) what type you obtained (for instance, written or verbal, and if verbal, how it was documented and witnessed). If your study included minors, state whether you obtained consent from parents or guardians. If the need for consent was waived by the ethics committee, please include this information.

2.Please amend your detailed Financial Disclosure statement. This is published with the article. It must therefore be completed in full sentences and contain the exact wording you wish to be published.

3. In the online submission form, you indicated that "The datasets used and analyzed during the present study are available from the corresponding author on reasonable requests". All PLOS journals now require all data underlying the findings described in their manuscript to be freely available to other researchers, either 1. In a public repository, 2. Within the manuscript itself, or 3. Uploaded as supplementary information.

Additional Editor Comments (if provided):

1 Add some evidences of diabetes-related self-care practices of Ethiopia in Introduction

2 How many public health institutions were included in this study?

3 The sampling fraction was: 815/392= 2.1. From where the authors used the number 292?

4 Need more information about data collection. Authors mainly described about the questionnaire.

5. Add a supplementary file of the questionnaire that used to collect data

6. Add a section of ‘Operational definition’ to define the key variables of interest

7. Table 3: Mean score of self-care practices in North East Ethiopia, 2021. How mean score was calculated is not clear. Describe clearly in the method section.

8. In the first paragraph of discussion, give a summary of objective-wise findings.

9. Discussion need to be re-write. Add more evidences from global research, compare, and mention the causes of differences in the findings in between them.

10. Need language and grammatical correction using professional

11. Also address the reviewers comment.

Reviewers' comments:

Reviewer's Responses to Questions

**Comments to the Author**

1. Does this manuscript meet PLOS Global Public Health’s publication criteria? Is the manuscript technically sound, and do the data support the conclusions? The manuscript must describe methodologically and ethically rigorous research with conclusions that are appropriately drawn based on the data presented.

Reviewer #1: Yes

Reviewer #2: Partly

2. Has the statistical analysis been performed appropriately and rigorously?

Reviewer #1: Yes

Reviewer #2: No

3. Have the authors made all data underlying the findings in their manuscript fully available (please refer to the Data Availability Statement at the start of the manuscript PDF file)?

Reviewer #1: Yes

Reviewer #2: Yes

4. Is the manuscript presented in an intelligible fashion and written in standard English?

Reviewer #1: Yes

Reviewer #2: Yes

5. Review Comments to the Author

Reviewer #1: Thanks to the author for their efforts, dedication and nice work. The topic of the study is very much essential to explore the present situation of self care practices of DM among people of LMICs like Ethiopia. I have found no major issues in the draft , though i will suggest the author to add an conceptual framework of the study. Along with that if they add another flow chart of their sample collection procedure, it will be eye catchy, but its not mandatory. Overall, this is a nice attempt.

Reviewer #2: Dear authors,

Thank you for the manuscript. I found major fundamental flaws in Introduction and Methodology section. The scope of self practice is not clear and is not supported by sufficient scientific literature. Methodology wise it is not scientifically sound and robust, and does not follow conceptual framework. Ethics is not clearly expressed. It needs a lot of effort to make it scientifically sound.

Best wishes.

6. PLOS authors have the option to publish the peer review history of their article (what does this mean?). If published, this will include your full peer review and any attached files.

**Do you want your identity to be public for this peer review?** For information about this choice, including consent withdrawal, please see our Privacy Policy.

Reviewer #1: No

Reviewer #2: No

---

## [Decision Letter · Decision Letter 1]

31 Oct 2023

PGPH-D-22-00339R1

Self-Care Practices and its Barriers among Diabetes Patients in North East Ethiopia: A facility-based cross-sectional study

Dear Dr. Asefa,

Thank you for submitting your manuscript to PLOS Global Public Health. After careful consideration, we feel that it has merit but does not fully meet PLOS Global Public Health’s publication criteria as it currently stands. Therefore, we invite you to submit a revised version of the manuscript that addresses the points raised during the review process.

Please address Reviewer#1 remaining concerns in the attached document.

Please revise the manuscript to address all the reviewer's comments in a point-by-point response in order to ensure it is meeting the journal's publication criteria. Please note that the revised manuscript will need to undergo further review, we thus cannot at this point anticipate the outcome of the evaluation process.

We look forward to receiving your revised manuscript.

Kind regards,

Miquel Vall-llosera Camps

Staff Editor

Journal Requirements:

1. Please provide a copy editing

Reviewers' comments:

Reviewer's Responses to Questions

**Comments to the Author**

1. If the authors have adequately addressed your comments raised in a previous round of review and you feel that this manuscript is now acceptable for publication, you may indicate that here to bypass the “Comments to the Author” section, enter your conflict of interest statement in the “Confidential to Editor” section, and submit your "Accept" recommendation.

Reviewer #1: All comments have been addressed

2. Does this manuscript meet PLOS Global Public Health’s publication criteria? Is the manuscript technically sound, and do the data support the conclusions? The manuscript must describe methodologically and ethically rigorous research with conclusions that are appropriately drawn based on the data presented.

Reviewer #1: Partly

3. Has the statistical analysis been performed appropriately and rigorously?

Reviewer #1: Yes

4. Have the authors made all data underlying the findings in their manuscript fully available (please refer to the Data Availability Statement at the start of the manuscript PDF file)?

Reviewer #1: No

5. Is the manuscript presented in an intelligible fashion and written in standard English?

Reviewer #1: No

6. Review Comments to the Author

Reviewer #1: The topic is quite important for the country context as well as the world context. But i have found many portions are still lack of clarification though it's a revision version. I have added my comments on track changes and hope it will be helpful for the authors.

7. PLOS authors have the option to publish the peer review history of their article (what does this mean?). If published, this will include your full peer review and any attached files.

**Do you want your identity to be public for this peer review?** For information about this choice, including consent withdrawal, please see our Privacy Policy.

Reviewer #1: **Yes: **Fardina Rahman Omi

---

## [Decision Letter · Decision Letter 2]

11 Jan 2024

Self-Care Practices and its Barriers among Diabetes Patients in North East Ethiopia: A facility-based cross-sectional study

PGPH-D-22-00339R2

Dear Dr. Asefa,

We are pleased to inform you that your manuscript 'Self-Care Practices and its Barriers among Diabetes Patients in North East Ethiopia: A facility-based cross-sectional study' has been provisionally accepted for publication in PLOS Global Public Health.

Best regards,

Zulkarnain Jaafar

Academic Editor

Reviewer Comments (if any, and for reference):

Reviewer's Responses to Questions

**Comments to the Author**

1. If the authors have adequately addressed your comments raised in a previous round of review and you feel that this manuscript is now acceptable for publication, you may indicate that here to bypass the “Comments to the Author” section, enter your conflict of interest statement in the “Confidential to Editor” section, and submit your "Accept" recommendation.

Reviewer #1: All comments have been addressed

Reviewer #3: All comments have been addressed

2. Does this manuscript meet PLOS Global Public Health’s publication criteria? Is the manuscript technically sound, and do the data support the conclusions? The manuscript must describe methodologically and ethically rigorous research with conclusions that are appropriately drawn based on the data presented.

Reviewer #1: Yes

Reviewer #3: Yes

3. Has the statistical analysis been performed appropriately and rigorously?

Reviewer #1: Yes

Reviewer #3: Yes

4. Have the authors made all data underlying the findings in their manuscript fully available (please refer to the Data Availability Statement at the start of the manuscript PDF file)?

Reviewer #1: Yes

Reviewer #3: Yes

5. Is the manuscript presented in an intelligible fashion and written in standard English?

Reviewer #1: Yes

Reviewer #3: Yes

6. Review Comments to the Author

Reviewer #1: (No Response)

Reviewer #3: The study is good in assessing self-care practices among diabetes patients. Going forward you could consider longitudinal studies which are more suited establishing causal relationships among various factors, considering all confounders of course.

7. PLOS authors have the option to publish the peer review history of their article (what does this mean?). If published, this will include your full peer review and any attached files.

**Do you want your identity to be public for this peer review?** For information about this choice, including consent withdrawal, please see our Privacy Policy.

Reviewer #1: **Yes: **Fardina Rahman Omi

Reviewer #3: **Yes: **Abdul-Basit Abdul-Samed
